# From Petri Dish to Field: Plant Tissue Culture and Genetic Engineering of Oats for Improved Agricultural Outcomes

**DOI:** 10.3390/plants12213782

**Published:** 2023-11-06

**Authors:** Krishna Mohan Pathi, Thorben Sprink

**Affiliations:** Julius Kuehn Institute (JKI)—Federal Research Centre for Cultivated Plants, Institute for Biosafety in Plant Biotechnology, 06484 Quedlinburg, Germany

**Keywords:** double haploids, somatic embryogenesis, protoplast technology, embryogenic callus, *Agrobacterium* transformation, biolistics, genome editing, oat biotechnology, wide hybridization, embryo-rescue

## Abstract

Oats (*Avena sativa*) hold immense economic and nutritional value as a versatile crop. They have long been recognized as an exceptional choice for human consumption and animal feed. Oats’ unique components, including proteins, starches, and β-glucans, have led to its widespread use in various food products such as bread, noodles, flakes, and milk. The popularity of oat milk as a vegan alternative to dairy milk has soared due to the increasing number of vegetarians/vegans and growing environmental awareness. Oat milk offers a sustainable option with reduced greenhouse gas emissions during its production, rendering it an appropriate choice for individuals who are lactose-intolerant or have dairy allergies. To ensure improved adaptability and enhanced nutrition, the development of new oat varieties is crucial, considering factors like cultivation, climate, and growing conditions. Plant cell culture plays a crucial role in both traditional and contemporary breeding methods. In classical breeding, plant cell culture facilitates the rapid production of double haploid plants, which can be employed to accelerate the breeding process. In modern breeding methods, it enables genetic manipulation and precise genome editing at the cellular level. This review delves into the importance of oats and their diverse applications, highlighting the advantages of plant cell culture in both classical and modern breeding methods. Specifically, it provides an overview of plant tissue culture, encompassing genetic transformation, haploid technology, protoplast technology, and genome editing.

## 1. Introduction

The common oat (*A. sativa*) is a temperate cereal grain from the *Poaceae* family that is widely cultivated worldwide. *Avena* species exhibit genetic diversity as they occur naturally as diploids, tetraploids, and hexaploids, which contributes to their genetic variability [1]. Cultivated oats are allohexaploid (AACCDD, 2n = 6x = 42) and are believed to have been domesticated over 3000 years ago. They were initially found as weeds in wheat, emmer, and barley fields in Anatolia [2,3]. Oats are a nutritionally significant crop with global importance, serving as a vital source of human and livestock feed. High levels of calcium and β-glucan soluble fiber in oats play a pivotal role in providing a wide range of health benefits [4,5,6] and high-quality oil and protein [7,8]. Furthermore, owing to their lack of gluten and low levels of gluten-related prolamins, oat seed represents a healthy substitute for gluten-intolerant individuals. Oats are rich in polyphenolic avenanthramides, which possess antioxidant, anti-inflammatory, and antiatherogenic properties [9]. Furthermore, oat grains encompass a tandem of saponin compounds, namely avenacosides (steroid-conjugated sugars) and avenacins (triterpenoid-conjugated sugars), which have been demonstrated for their capacity to reduce cholesterol levels enhance immune system activity and exhibit anti-carcinogenic attributes [7]. 

Recently, there has been a significant surge in the popularity of oat milk due to a multitude of factors including a growing number of vegetarian and vegan people and a growing understanding of our behavior for global warming. Oat milk is a high-quality product that offers a palatable vegan substitute for traditional dairy milk. Not only does oat milk provide a sustainable alternative to dairy milk, but it also emits fewer greenhouse gases during its production as oat is cultivated locally in most parts of the world, especially in Europe and northern America, thereby benefiting the environment (Table 1). Furthermore, oat milk caters to the dietary needs of individuals who are lactose intolerant or allergic to dairy products. Consequently, oat milk represents a promising market opportunity for food and beverage companies [10]. 

The advent of climate change necessitates an urgent requirement to develop novel oat varieties that can withstand the challenges posed by changing environmental conditions. The primary objective of breeding such oat varieties is to enhance their capacity to withstand drought, heat stress, pests, and other factors that are expected to intensify with climate change. In recent years, there has been a shortcoming of raw oats in Europe due to the increasing demand for fodder as well as for human consumption; especially, high quality oats have been short due to increased fungi pressure. Furthermore, oat can be used for the production of milk substitutes, but the protein content of oats is currently not high enough to use oats without additional proteins, for cheese or meat substitutes. However, this would be of high interest as oat-based substitutes are highly accepted and are in favor compared to other plant-based substitutes. Developing an oat-based meat substitute would hold significant value, particularly in circumventing the use of gluten.

Scientists are investigating diverse methods to create oat varieties that can acclimate to changing environmental conditions without compromising their nutritional value and quality. New genomic techniques have revolutionized the field of plant biotechnology, offering several advantages. One of the main advantages of these techniques is their precision and speed in introducing specific traits into the plant genome. Through gene editing, a precision-oriented approach, it becomes feasible to introduce specific alterations to the plant’s DNA sequence, thereby enabling targeted enhancements of favorable traits. Plant cell culture is an essential tool in both classical and modern breeding techniques. In classical breeding, plant cell culture facilitates the rapid production of double haploid plants, which can be employed to accelerate the breeding process. This process streamlines the identification of desirable traits, leading to a more efficient and accurate selection of plant varieties with enhanced genetic characteristics. In modern breeding, plant cell culture provides a potent means of genetic manipulation and desired trait propagation. Within the context of genome editing, this approach permits the precise manipulation of plant cells at the cellular level, allowing for targeted modifications to be introduced into the plant’s DNA. Furthermore, plant cell culture offers a mechanism for the proliferation of modified cells, facilitating the generation of a substantial quantity of plants possessing the desired genetic traits. 

To our current understanding, this is the foremost comprehensive review elucidating the in vitro and biotechnological aspects of oats. Our analysis encompasses a comprehensive overview of plant tissue culture, genetic transformation, protoplast technology, double haploid technology, wide hybridization, and genetic engineering.

### Plant Cell Culture

In vitro cultivation of plant cells offers a multitude of versatile applications, with the added benefit of the totipotent nature of plant cells amplifying their potential in the realm of plant biotechnology. Plant cell culture, which originated in the early 1960s, has become a cornerstone of modern biotechnology. 

Mass propagation: plant tissue culture enables fast and large-scale propagation of plants, which is beneficial in meeting the high demand for plants in commercial production [12,13]. Rapid multiplication: it enables the quick multiplication of plants through clonal propagation, in which a small piece of plant tissue can generate numerous identical plants [14,15]. Disease-free plants: it produces disease-free plants that are crucial for developing new cultivars in plant breeding by eliminating pests, contaminants, and pathogens [16,17]. Genetic manipulation: it enables the genetic manipulation of plants, leading to the development of novel cultivars endowed with specific traits [18,19]. Conservation: it plays a pivotal role in safeguarding rare and endangered plant species by offering an in vitro preservation method, which allows for their protection and eventual reintroduction into their native habitat [20,21]. Reduced space requirements: it is an ideal option for small-scale environments like laboratories since it requires less physical space compared to traditional plant propagation techniques. Control over growth conditions: it provides precise control over growth conditions like temperature, light, and nutrient levels, enabling cultivation of plants under conditions that are difficult to achieve in the natural environment. Molecular forming/metabolic engineering of fine chemicals: it can be used for molecular farming, producing valuable complex molecules or recombinant proteins using plant cells, with potential applications in agriculture, industry, and medicine [22]. Better understanding: it provides a controlled experimental platform for researchers to study cellular processes and responses under various environmental and genetic conditions, making it an important model system for investigating fundamental aspects of plant cell physiology.

## 2. Oat Tissue Culture 

Numerous endeavors were undertaken to establish the effective utilization of plant tissue culture (Table 2) and genetic transformation techniques in the cultivation of oats. Kaufman [23] conducted a study on the impact of Indole-3-Acetic Acid (IAA) on cell division, and cell enlargement within the intercalary meristem of *Avena* internodes. Later, Webster [24] successfully instigated the emergence of callus from germinating whole seedlings of hulled oats. The exclusive formation of callus was observed only when IAA, 1-Naphthaleneacetic acid (NAA), and 2,4-dichlorophenoxyacetic acid (2,4-D) were present. A firm and golden callus was produced within 6 weeks of inoculation in a medium containing 2,4-D, IAA, and glucose. The callus was maintained through several sub-cultures for 3 years. Afterwards, Carter et al. [25] demonstrated successful callus induction from the root system in the presence of an appropriate concentration of auxin in the medium. Subsequently, Brenneman and Galston [26] induced callus from *A. sativa* hypocotyls on high auxin medium, which was subsequently subcultured and induced roots on media lacking auxin or enriched with cytokinins. Modifying salt concentrations in the medium led to the formation of green nodules with meristemoids, but no shoots were produced. After that, Cummings et al. [27] achieved successful regeneration in multiple genotypes by initiating callus from immature embryos. Moreover, the investigation also took into account the possibility of using excised apical meristem tissues and germinating mature embryos as viable sources of calli with the capacity for plant regeneration. Subsequently, Rines and McCoy [28] achieved successful plant regeneration from tissue cultures of three hexaploid oat species, encompassing the cultivated oat (*A. sativa*) as well as two wild oat varieties (*A. sterilis* L. and *A. Jatua* L.), from immature embryos. Regenerable-type cultures were characterized by organized chlorophyllous primordia in a compact, yellowish-white, highly lobed callus. The frequency of regenerable-type cultures was influenced by factors such as embryo size, species, genotype, and growth conditions. Next, Nabors et al. [29] described a technique for generating a callus from oat seedling roots that can lead to shoot regeneration. The occurrence of green spots in the secondary callus is positively correlated with shoot regeneration and requires auxin for continued growth. Additionally, it was observed that a high-temperature regime (30 °C) is necessary for the establishment of regenerated plants under greenhouse or growth chamber conditions, whereas cooler temperatures (20 °C) are required for seed set. In a follow-up study Heyser and Nabors [30] reported the production of both embryogenic (white and opaque) and non-embryogenic calli (rough and yellow) as well as green spots in oat using mature seeds, mesocotyls, and immature embryos. Both types of callus were capable of producing shoots and roots, but the embryogenic callus produced complete plantlets at higher frequencies. Shekhawat et al. [31] reported a technique for initiating totipotent cultures from young secondary tillers of 60 to 75-day-old plants using small explants. Tiller-bud cultures require less work to initiate compared to cultures started from immature embryos. Rines and Luke [32] investigated genetic and chromosomal changes induced by culture conditions and chemical agents in tissue cultures. The researchers selected for insensitivity to the *Helminthosporium victoriae*-produced pathotoxin victorin in tissue cultures of oat lines carrying the sensitive allele Vb. Sixteen cultures grown on toxin-containing medium produced surviving callus sectors or shoots, and nine of these lines produced toxin-insensitive plants. These plants coincidentally lost the Vb crown rust resistance, but no chromosomal deficiency was identified to explain the loss of toxin sensitivity. Bregitzer et al. [33] describe the development and characterization of oat callus, focusing on the creation of friable, embryogenic callus and its subsequent properties. The authors isolated embryogenic sectors from the callus and repeatedly subcultured them to generate a friable embryogenic callus, which was then matured into somatic embryos and transferred to a hormone-free medium to allow for germination. The researchers regenerated plants from the callus lines for more than 78 weeks and observed genotypic variation in the response to embryogenic callus initiation. Chen et al. [34] induced callus and plant regeneration from six genotypes of oat leaves. A regenerable callus was induced in the basal 1-2 mm region of seedlings aged between 2 to 5 days, and subsequently, the regenerated plants were cultivated and allowed to mature. The regenerative capacity of the first 1 mm of the leaf basal region from three-day-old seedlings was similar to that of immature embryos. Moreover, the calli capable of leaf regeneration were able to maintain their ability to regenerate even after being subcultured for eight consecutive months. Similarly, Chen et al. [35] developed an oat plant regeneration system using leaf tissue from seedlings. The callus derived from the leaf base had a higher response of plant regeneration. Somatic embryogenesis was observed from the callus near the apical meristem. The optimal regeneration frequency of 60% was achieved using various auxin concentrations in plant regeneration media. The study also investigated the effects of donor plant age and hormones on regeneration. Zhang et al. [36] developed an in vitro plant regeneration system using the shoot apical meristem of four oat cultivars. Multiple shoots were induced for all cultivars using different hormone combinations in an MS basal medium, and fertile oat plants were produced from multiple shoots. Gless et al. [37] developed a regeneration protocol for oat plants using leaf base segments of young in vitro seedlings. The study observed the effect of the developmental stage and genotype on callus induction and regeneration efficiency for five genotypes. The oat leaf bases demonstrated high regeneration potential, with an average of 25 plants per explant and up to 50 regenerants per explant for the most responsive genotype. Later, A. Birsin [38] investigated callus induction and regeneration capacity in ten oat genotypes using mature embryos. Results showed that both traits varied among genotypes, but no significant correlation was found between the callus induction and plant regeneration capacity. Kelley et al. [39] investigated the regenerability of seven oat genotypes using three tissue culture methods: callus, multiple meristem, and multiple meristem-to-callus combinations. The results showed that GP-l produced significantly higher plant numbers than the other genotypes. The study also noted that when GP-l served as the maternal parent, the genotypes GP-l × Corbit and Corbit × GP-l demonstrated a notable increase in plant numbers during their production. M Jung-Hun [40] successfully regenerated Korean oat accessions (Malgwiri and Samhangwiri) from mature embryos and leaf base segments. A callus induction medium supplemented with 3 mg/L of 2,4-D and 3 mg/L of picloram showed a high regeneration frequency, particularly for Malgwiri with a regeneration efficiency of 74%. Additionally, the regeneration media containing 1 mg/L of 2,3,5-triiodobenzoic acid (TIBA) and 1 mg/L of Benzyl Adenine (BA) increased the frequency of multiple shoots while reducing the shoot initiation period. Borji et al. [41] used environmental scanning electron microscopy and transmission electron microscopy to observe the micromorphological and structural changes during oat somatic embryogenesis. They identified different organelles involved in the embryogenic cells and observed several stages of the process, including somatic embryo germination. Salunke et al. [42] induced callus and regenerated plants from JO-1 and OS-6 oat accessions using mature and immature embryos, as well as leaf bases as explants. Al Mamun et al. [43] reported on the in vitro response of oat callus induction under NaCl salt stress and subsequent plant regeneration. 


**Genetic Transformation of Oats**


A series of investigations were undertaken to establish the genetic transformation in oats (Table 3). Somers et al. [44] reported a successful generation of transgenic oat plants using a biolistic gene transfer method. The *BAR* (*Phosphinothricin*) and *Escherichia coli uidA* (*GUS*) genes were utilized as selection markers, with the former providing herbicide resistance in the plants. Out of the 111 transgenic tissue cultures, 38 gave rise to regenerated plants, most of which displayed male sterility, while over 30 were fully fertile. The seeds of the fertile plants exhibited stable inheritance of the transgenes, demonstrated by GUS activity and PPT resistance. Later, Gless et al. [45] developed transgenic oat plants using freshly isolated leaf base segments bombarded with plasmids containing the *UIDA* and *PAT* genes. A 5% frequency was observed, resulting in the recovery of transgenic plantlets. The integration of foreign genes was confirmed through Southern blot analysis. These plants appeared normal and were mostly fertile, demonstrating Mendelian inheritance of the introduced genes to the next generation. Further, Zhang et al. [46] transformed oat using shoot meristematic cultures. After selection, they obtained seven independent transgenic lines, five of which were self-fertile. The transgenic oat exhibited both Mendelian and non-Mendelian segregation ratios of transgene expression in the T1 and T2 progeny, as well as both normal and low physical transmission of the transgenes. Afterwards, Gasparis et al. [47] genetically transformed three Polish spring oat cultivars (Bajka, Slawko, and Akt) through *Agrobacterium*-mediated transformation. The highest transformation rate was achieved in immature embryos of cv. Bajka, but only a fraction of T0 and T1 plants showed *GUS* expression. Southern blot analysis showed a simple integration pattern with a low copy number of introduced transgenes. Dattgonde et al. [48] also described an *Agrobacterium*-mediated transformation system and tested different co-cultivation treatments with varying incubation periods. The most effective method was vacuum infiltration with a 72 h dark incubation.

## 3. Protoplast Technology

A plant protoplast refers to a plant cell that has had its cell wall removed through enzymatic or mechanical means, leaving only the plasma membrane and its associated organelles intact. Protoplast technology has several advantages in plant research and breeding [51]. Facilitates genetic manipulation: The removal of the cell wall in protoplasts facilitates the introduction of DNA and proteins (ribonucleoproteins (RNPs), into plant cells with relative ease [19]. This allows for the creation of genetic manipulations. Enables hybridization: Protoplast fusion technology results in the formation of hybrid cells that combine traits from distinct parent plants. This innovative approach enables the creation of new plant varieties possessing desirable characteristics [52,53,54]. Allows for rapid screening: protoplasts can be screened for desirable traits quickly and efficiently, allowing plant breeders to identify potential new plant varieties more rapidly [55]. Facilitates the study of cell physiology: protoplasts aid in studying plant cell physiology by eliminating the cell wall barrier, facilitating the examination of cellular processes within the cell [56]. Regeneration from a single cell: this ability to regenerate plants from a single cell has significant implications for plant breeding and genetic engineering, as it allows for the creation of genetically identical plant populations with desirable traits [57]. Manipulations at single cell level: plant protoplast technology offers the possibility of genetic manipulation at the single-cell level [57]. 

Various studies have been carried out to examine the techniques employed for the isolation of protoplasts, as well as their subsequent response to plant growth regulators. Brenneman and Galston [26] successfully isolated viable protoplasts from *A. sativa* leaves and observed their behavior on defined agar and liquid media, revealing that cell division occurred unpredictably and sporadically and was not perpetuated. The study also found that biotin induced a greater number of multiple cell clusters, while coconut milk was harmful to the culture, and that the growth of protoplasts was affected by factors such as incubation temperature, light, and photoperiods. Later, Fuchs and Galston [58] investigated the uptake of labeled leucine, uridine, and thymidine in oat leaf protoplasts. Optimal incorporation rates were observed with 0.6 M Mannitol, whereas glucose or inositol led to a reduction in leucine incorporation. Additionally, Cycloheximide and kinetin were found to inhibit L-leucine incorporation. Interestingly, no effect was observed with regard to the incorporation of L-leucine or uridine in response to treatment with auxins, abscisic acid, or gibberellic acid. Further, Kaur-Sawhney et al. [59] found that pretreatment with cycloheximide and kinetin increased the yield and metabolic activity of oat leaf protoplasts, while enhancing their resistance to spontaneous lysis. Following that, Altman et al. [60] found that natural polyamines stabilize leaf protoplasts against lysis, preventing a senescence-induced reduction in RNA and protein synthesis and increase in RNase activity. Eastwood [61] isolated spheroplasts from oat aleurone layer cells and observed their disorganization in low D-mannitol solutions. The study concluded that the requirement for an osmoticum limits the utility of spheroplasts as a model for investigating gibberellin mechanisms in the aleurone cell. Thereafter, Kaur-Sawhney et al. [62] found that oat leaf protoplasts had high nuclease activity and low macromolecule incorporation, which led to spontaneous lysis after 18 h of buffer floating. They discovered that adding senescence retardants like cydoheximide or kinetin, dibasic amino acids like L-lysine or L-arginine, or diamines such as putrescine or cadaverine reduced nuclease activity and spontaneous lysis, while improving protein and nucleic acid synthesis rates. Diamines also delayed chlorophyll degradation and improved the quality of the excised leaves. However, the senescence promoter L-serine had the opposite effect, accelerating chlorophyll degradation and not enhancing protoplast quality. Later, Galston et al. [63] found that senescence-retarding diamines, specifically L-arginine and L-lysine, could stabilize oat leaf protoplasts by regulating endogenous RNAase levels. These amino acids delayed spontaneous lysis, prevented aggregation and adhesion, and maintained uniform chloroplast distribution. Furthermore, they offered protection against stressors that could induce protoplast lysis, such as osmotic shock, exogenous RNAase, and cell-free centrifugal supernatant fractions of mechanically lysed protoplasts. Subsequently, Kaur-Sawhney et al. [64] found that incorporating polyamines and dibasic amino acids into the isolation medium decreased RNase activity in extracted oat leaf protoplasts compared to controls. Polyamines had a stronger inhibitory effect than dibasic amino acids, likely due to their electrovalent attachment to RNA, and the inhibitory effect was correlated with their positive charge. Dibasic amino acids were thought to act by converting to polyamines during protoplast isolation. Latterly, Hooley [65] isolated viable protoplasts from mature wild oat aleurone layers, yielding over 90% of aleurone layer cells. The protoplasts were responsive to GA3 treatment and underwent vacuolation during in vitro incubation, which was stimulated by GA3, though not exclusively dependent on it. After that, Hahne et al. [66] showed that oat mesophyll protoplasts can be dedifferentiated and induced to form colonies using specific plant growth regulators and modified culture conditions, with effectiveness varying across oat genotypes. Later, Rakotondrafara et al. [67] presented a detailed protocol for successful electroporation of oat protoplasts obtained from cell suspension culture.

## 4. Double Haploid Technology

Doubled haploids (DH) are a unique and valuable plant material with numerous applications in plant breeding and research. DH plants are produced through haploidization induction, followed by chromosome doubling, resulting in complete homozygosity for all chromosomes. This phenomenon facilitates the generation of genetically uniform lines within a shorter timeframe compared to conventional breeding approaches. The resulting homozygosity of DH plants provides an unequivocal display of genotypes by phenotype, making them useful for genetic analysis and trait evaluation. Additionally, each individual DH plant derives from a random meiotic recombination, making them highly diverse and useful in breeding programs. DH plants have been used as breeding lines for crosses and as parental lines for hybrid breeding. They can also be used in mapping populations and in the development of near-isogenic lines, where they provide a valuable resource for genetic and phenotypic comparison. 

The utilization of DH technology confers numerous benefits in contemporary breeding methodologies. One such application, genetic transformation, whereby genes are transferred to haploid target cells, is of great scientific interest due to the ability to integrate transgenes and double the genome, which provides a unique opportunity to immediately generate plants that are homozygous for the transferred gene [68,69]. Moreover, the application of RNA-guided endonucleases (RGENs) to haploid cells enables the direct production of homozygous mutant plants, thereby expanding the potential applications of this technique [70]. Furthermore, the principle of haploid induction (either with wide crosses or haploidy inducers), coupled with site-directed mutagenesis, presents a promising strategy for generating mutated plants that are devoid of T-DNA and exhibit a high degree of homozygosity. Such a methodology has the capacity to mitigate the influence of genotype on site-directed mutagenesis outcomes [71,72].

Several investigations have aimed to establish the employment of doubled haploid (DH) technology in oats by utilizing anther and microspore culture techniques (Table 4). Chung [73] documented the production of calli via anther culture, and subsequently, the microspore-derived nature of these calli was verified by means of histological analysis of successive sections of an anther-derived callus. Later, Rines and McCoy [74] initiated callus cultures from anthers containing microspores, but the attempt failed to yield any haploid plantlets. In a subsequent study, Rines [75] reported the production of a haploid plant through anther culture in oats, although efficiencies were very low. Only a single haploid plant and two diploid plants were retrieved from a total of over 65,000 oat anthers, with 2627 anthers generating calli. Further, Sun Jing-san [76] achieved the successful regeneration of haploid oat plants via anther culture. Although the production of haploid oat plants through anther culture is a viable method, it remains restricted by the genotype and a low frequency of success. Afterwards, Kiviharju et al. [77] reported the first successful anther culture in the wild, hexaploid, highly self-pollinating oat species *A. sterilis*. The researchers confirmed that this specific oat species consistently produced embryogenic structures, which have the ability to regenerate both green and albino plantlets. In a follow-up study, Kiviharju and Pehu [78] studied stress pretreatments’ impact on embryo induction in anther cultures of oat species genotypes. They found that the *A. sativa* line WW 18019 and *A. sterilis* line CAV 2648 responded best to a heat pretreatment. The study also showed that maltose was a more effective carbon source than sucrose, with 14% maltose being the most effective in inducing embryos in *A. sterilis.* Furthermore, Kiviharju et al. [79] tested anther culture ability in different oat genotypes and found that naked and wild oat genotypes produced more embryos than other oat genotypes. The study revealed that only wild oat genotypes were capable of plant regeneration, and this ability was transmitted to the offspring of the *A. sativa* × *A. sterilis* cross cv. Puhti × CAV 2648. Furthermore, the response of oat anthers to regeneration was suppressed by auxin, while naked oat, wild oat, and *A. sativa* × *A. sterilis* crosses demonstrated greater responsiveness to a medium supplemented with 2,4-D. Moreover, Kiviharju and Tauriainen [80] investigated the effects of 2,4-D and kinetin on anther culture of *A. sativa* L. (cultivated oats), wild oats (*A. sterilis* L.), and their progeny. The results revealed that a high concentration of 2,4-D enhanced embryo production in both species, and also promoted plant regeneration in *A. sterilis* and *A. sativa A. sterilis* progeny. In contrast, the administration of kinetin resulted in severe browning. However, a low concentration of kinetin was found to be crucial for regenerable embryos in *A. sativa* cv. Kolbu. In addition to this, Kiviharju et al. [81] induced regenerable-type embryos from heat-pretreated anthers of various oat genotypes and studied the effects of growth regulators and basic medium on plant regeneration. The study found that 6-Benzylaminopurine (BAP) promoted albino plant regeneration, particularly in W14 medium. Of the 14 genotypes tested, only 4 produced green plants, while 2 produced only albinos. Later, the same authors, Kiviharju et al. [82], reported significantly high plant regeneration rates from two oat cultivars, with 30 green plants per 100 anthers and over 500 regenerants in total. These numbers were deemed acceptable for DH production, serving cultivar breeding and genetic study purposes. Ślusarkiewicz-Jarzina and Ponitka [83] investigated the effect of different physical-state media on androgenic response and plant regeneration in oat hybrids. They obtained embryo-like structures from eight genotypes and successfully regenerated plants from two. The highest frequency of embryo induction occurred in one genotype on a solid medium, from which 35 plants were regenerated. In a subsequent investigation, Ponitka and Ślusarkiewicz-Jarzina [84] conducted a study to assess the impacts of C17 and W14 induction media on the development of embryo-like structures (ELS) in nine hexaploid oat hybrids. C17 was more effective, and the highest rate of green plants was achieved at 22 °C in the dark. Of the 36 green regenerants obtained, 28 were haploid, 8 were spontaneous doubled haploids, and 19 were partially fertile after colchicine treatment. Ferrie et al. [85] documented the successful generation of green double haploid (DH) plants with minimal albinism via microspore cultures in oats. High microspore densities leading to effective embryogenesis and colchicine treatment contributed to the conversion of 80% of plants from haploid to DH. The study also examined the effects of mannitol pretreatment.

Warchoł et al. [86] conducted a study on anther culture to produce embryo-like structures and doubled haploid plants in four oat cultivars-Akt, Bingo, Bajka, and Chwat. Despite oat’s androgenesis resistance, all cultivars produced embryo-like structures. The Akt and Chwat cultivars, in particular, produced fertile double haploid plants. In their study, the researchers demonstrated that an enhanced androgenic response could be attained by employing a combination of cold pretreatment and heat treatment on cut oat tillers before isolating anthers. They also utilized a specific medium with precise concentrations of 2,4-D and kinetin for induction. Furthermore, their findings suggested that the androgenic response was influenced by the distance between the base of the flag leaf and the penultimate leaf of the panicle. In addition to this, Warchoł et al. [87] studied the effect of Zinc, Copper, and Silver Ions combined with cold treatment on androgenesis in oat. The study found that adjusting the concentrations of copper, zinc, and silver in the media can promote androgenesis in oat, with the highest number of embryo-like structures observed on anthers treated with CuSO_4_ during pretreatment or when AgNO_3_ is added to the induction medium.


**Wide Hybridization**


Wide hybridization is a breeding technique in which two parents from different species or even genera are crossed. Due to this cross, the resultant embryo may encounter developmental failure caused by the chromosomal elimination of one of the parental genomes during the early stages of embryonic divisions. However, the embryo can be rescued and cultivated in vitro to produce haploid plants. Subsequent treatment with chemicals or spontaneous genome duplication can result in the production of double haploid plants. Wide hybridization has been successfully used to produce double haploid plants in various plant species. For example, haploid wheat plants have been produced by crossing wheat with maize, while potato haploids have been obtained through crossing with tomato. Barley haploids have been achieved through crossing with wheat and Brassica napus haploids through crossing with Sinapis arvensis. Moreover, *Capsicum annuum* haploids can be obtained through crossing with *C. chinense*, *Citrus reticulata* haploids through crossing with *Poncirus trifoliata*, and *Oryza sativa* haploids through crossing with *O. officinalis*, a wild rice relative. Wide hybridization is a valuable tool for the production of double haploid plants in plant breeding programs. 

Haploid individuals of cultivated oat (*A. sativa* L., 2n = 6x = 42) have been successfully produced through wide hybridizations with Panicoidae species, mainly with maize (*Zea mays*). The maize wide cross method for oat haploid production has been found to be less genotype restricted than the anther culture method. However, the plant recovery frequencies reported thus far for oat haploids produced through maize pollination are much lower. 

It has been observed that oat × maize hybridizations sometimes lead to the retention of maize chromosomes in the recovered oat plants. Additionally, these hybridizations can result in partial self-fertility in oat haploid plants.

Researchers made significant efforts to produce double haploids through wide hybridizations. The success of generating haploids from the hybridization of wheat and maize served as a driving force for researchers exploring this method. First, Rines and Dahleen [88] successfully generated oat haploid plants through oat × maize crosses. They applied maize pollen on emasculated oat florets and retrieved 14 haploid oat plants from 3300 emasculated florets. These haploid plants had 21 oat haploid chromosomes in their root tip cells but exhibited abnormal chromosome behavior and numerous micronuclei in their meiotic cells. Notably, the process was not genotype-dependent, and the haploid plants sometimes produced seeds, giving rise to both euploid and aneuploid progeny. Machan et al. [89] conducted crossings between oats (*A. sativa*) and both maize (*Zea mays*), teosinte (*Zea mays* ssp. *Mexicana*). Results showed that crossing oats with maize yielded a 4.52% embryo formation rate, while crossing oats with teosinte did not result in any embryo formation. A total of 24 plants were successfully obtained from oat × maize crosses. Matzk [90] demonstrated the production of haploid plants in oats via crosses with pearl millet, eastern gamagrass, and maize. Embryo frequencies ranged from 0.4% to 9.8%, depending on the pollinator species used. Notably, the pollinator species’ chromosomes were lost in later developmental stages, with only one to four remaining in root tip cells during the tillering stage. Four viable plants were produced, including hybrids with pearl millet and eastern gamagrass. Later, Riera-Lizarazu et al. [91] reported the cytological and molecular characteristics of oat × maize progenies. The hybrids contained 1-4 maize chromosomes and partial self-fertility in oat haploid plants. RFLP analysis identified the maize chromosomes present, and selfing led to transmission of the maize chromosome, resulting in self-fertile oat plants with an additional chromosome or chromosome pair. Afterwards, Sidhu et al. [92] examined the effects of the plant genotype, growth regulator application, temperature, and dicamba concentration on oat haploid plant production. Dicamba had the greatest impact on caryopsis production, followed by picloram, 2,4-D, and gibberellic acid. Temperature also had a significant effect on caryopsis and embryo production, which was genotype-dependent. The study observed haploid embryo production rates ranging from 0.8% to 6.7%, with 72% to 81% of haploids surviving and successfully doubling. Later, Ishii et al. [93] conducted a study where pearl millet pollen was utilized to fertilize the pistils of different plants within the Triticeae tribe and oat. The resulting embryos obtained from oat crosses retained all seven chromosomes of pearl millet. Thereafter, Marcińska et al. [94] delved into the impact of multiple variables on oat haploid generation, such as genotype, pollination timing, growth regulators, and the time at which they were applied. The findings of the study indicated that the oat genotype played a crucial role in both the frequency of haploid embryo formation and the success rate of plant regeneration. The most effective pollination occurred on the 2nd day after emasculation, and treating the florets with auxin on the 2nd day after pollination yielded better results compared to treatments applied after 1 or 3 to 5 days. Haploid embryos were obtained from all genotypes, and no albino plants were observed. Further, Noga et al. [95] reported that, the success of oat haploid embryo germination was found to be influenced by both the developmental stage of the embryos and the specific type of growth regulators incorporated into the regeneration medium. Then, Skrzypek et al. [96] found that a higher light intensity during in vitro culture increased embryo germination, plant conversion, and DH line production in oat. The study also emphasized the noteworthy influence of growth conditions of the donor plant on the development of haploid embryos. Subsequently, Kapoor and Singh [97] found that the frequency of embryo formation varied among oat genotypes, and the time interval between emasculation and pollination strongly influenced embryo production. Dicamba had a significant impact on the number of enlarged ovaries and embryos, with the highest number of embryos obtained when treating oat × maize florets with auxins after 2 days of pollination. Following this, Dziurka et al. [98] investigated the role of phytohormones in oat haploidization. Ovaries with developed embryos had higher levels of IAA, trans-zeatin (tZ), and kinetin than those without. Endogenous kinetin was negatively correlated with haploid embryo formation and tZ was negatively correlated with haploid plant growth in vitro, while GA1 had a positive correlation. IAA and tZ had a positive correlation in ovaries with developed embryos and a negative correlation in those without. Later, Juzoń et al. [99] evaluated 29 oat genotypes with different 2,4-D concentrations. Higher 2,4-D levels resulted in more haploid plants and DH lines with increased seed production. However, the effectiveness of the method may be limited due to genotypic variability. Next, Dziurka et al. [100] investigated the low germination rate of *A. sativa* haploid embryos resulting from distant crossing with maize. Morphological and anatomical differences were observed in haploid embryos of *A. sativa* compared to zygotic embryos, which may be due to significantly lower levels of endogenous auxins. Haploid embryos showed higher levels of cytokinins and a higher cytokinin to auxin ratio, indicating an earlier developmental stage. Gibberellins were found to be elevated in haploid embryos of the ‘Akt’ variety, whereas no such increase was observed in ‘Krezus’ embryos. The limited germination success of oat haploid embryos might also be attributed to an excessive production of reactive oxygen species and reduced levels of low-molecular-weight antioxidants and stress hormones.

## 5. Genome Editing in Oats

Site-directed mutagenesis represents a biotechnological methodology that deliberately modifies the DNA sequence at a predetermined position in the host genome through nucleotide insertion, deletion, or replacement. This highly efficient, flexible, and reliable technique is employed to rapidly generate new plant varieties harboring enhanced gene variants and traits. Moreover, these techniques enable the possibility of studying gene function and regulation, creating a significant impact on basic science. There are four primary classes of site-specific nucleases (SSNs): meganucleases, zinc finger nucleases (ZFNs), transcription activator-like effector nucleases (TALENs), and clustered regularly interspaced short palindromic repeat (CRISPR)-associated (Cas) endonucleases [101]. These aforementioned endonucleases can be customized to target a specific DNA sequence motif within live cells, and subsequently, the cellular DNA repair machinery processes the cleaved DNA. The mechanisms of cellular DNA repair include non-homologous end-joining (NHEJ) and homology-directed repair (HDR) [102] 


**CRISPR**


CRISPR genome-editing technology offers exciting opportunities for engineering desirable traits in plants with precise genome engineering and without the need for transgene integration. The CRISPR/Cas platform is based on RNA-guided Cas endonucleases, which are derived from the adaptive immune system of microbes. The Cas endonuclease system consists of two crucial constituents: a synthetic guide RNA (gRNA) and the Cas protein. The gRNA is designed to bind specifically to a user-defined DNA sequence, guiding the Cas9 endonuclease towards the target site for cleavage. The recognition of a protospacer motif, consisting of approximately 20 nucleotides, is achieved via complementary base pairing, allowing for the production of gRNAs for any desired sequence. The target motif also includes a protospacer-adjacent motif (PAM), comprising a few nucleobases downstream of the target motif, which binds to the Cas9 protein. The double-strand break (DSB) occurs between the third and fourth nucleotides in the 5′-direction from the PAM. After the DNA is cleaved, the cell’s inherent DNA repair mechanisms are triggered to initiate the repair process, employing either non-homologous end-joining or homology-directed repair pathways [55].

The use of CRISPR/Cas technology has demonstrated success in mono- and dicotyledonous plants through the implementation of single guide RNA (gRNA) expression systems. Examples of plant species in which this technology has been implemented include maize [103], barley [104], wheat [71], rice [105], tobacco [106], carrot [107], and chicory [19]. When a single cleavage site is targeted, it usually leads to short deletions and/or insertions. However, when pairs of target motifs are simultaneously targeted, it can result in significantly larger and precisely predictable deletions. In addition, targeting more than one genomic target site simultaneously can result in the deletion of large fragments [108,109] and can extend to encompass entire genes and chromosomal regions [110]. Multiplex genome editing has been demonstrated in plants through various studies [97,110,111,112,113,114]. Despite these successes, current utilization of Cas endonuclease technology in plants is still mainly limited to random mutagenesis caused by non-homologous end-joining (NHEJ)-based repair mechanisms. The targeted insertion or exchange of genes using homology-directed repair (HDR) has only been demonstrated in a limited number of situations, including model plants such as *Arabidopsis* [115] and *N. benthamiana* [108], as well as crops like soybean [116] and rice [117].

The integration of CRISPR cassettes into genomes is often non-specific and can cause regulatory and safety concerns. Breeding to segregate transgene elements from desired editing events may be impractical for species with long juvenile growth periods or that are vegetatively propagated. To avoid foreign DNA integration during genome editing, ribonucleoproteins (RNPs) are formed by preassembling the Cas protein and gRNA(s) of the CRISPR system and introducing them into plants. Edited plants can be considered transgene-free as no recombinant DNA is involved in this process. RNPs can be effectively utilized in all organisms without encountering delivery barriers. Additionally, there is no need to consider promoter compatibility or multiplexing strategies. Utilizing RNP-based CRISPR technology has the potential to produce improved germplasm that is transgene-free, making it easier to commercialize [118].

The engineering of oats using site-specific nucleases has not been extensively pursued. The first report on this subject emerged from Donoso [119], who studied the role of Thaumatin-like protein 8 (TLP8) using the CRISPR/Cas9 system. Three constructs were meticulously devised, with each one targeting a specific homeolog, namely AsTLP8 A, C, and D. Park lines were selected for transformation via microprojectile bombardment. Notably, these constructs exhibited transformation frequencies of 5.23%, 0.47%, and 2.86%, respectively. This research delves into the discovery of the inverse correlation between TLP8 expression and beta-glucan content in germinating barley seeds. This finding prompted an investigation into the regulatory role of TLP8 in beta-glucan synthesis in oats. Notably, the study demonstrates the successful editing of *AsTLP8* homeologs in oats using CRISPR/Cas9. However, the precise impact of these mutations on beta-glucan content requires further investigation.

## 6. Potential Challenges

Despite considerable endeavors to establish tissue culture and transformation protocols in oats, genotype dependency remains a prominent challenge. Oat regeneration protocols have been developed based on few genotypes such as the GAF and Park lines, and it is important to note that GAF and Park exhibit poor agronomic characteristics [32,44]. The most suitable material for oat transformation to enhance traits would be an elite variety possessing outstanding agronomic characteristics. However, successful transformations also depend on specific genotypes. For instance, biolistic transformations were demonstrated in GAF and Park lines [44]. While most plants displayed male sterility, subsequent attempts by other researchers achieved success with low transformation efficiency. Reports on *Agrobacterium*-mediated transformation in oats remain scarce. Thus, it is imperative to establish robust transformation protocols for this species. Protoplast isolation in oats succeeded, but no reports of callus formation or plant regeneration were found. This underscores the challenge in this aspect. Identifying the suitable genotype, cell types, and conditions for cell division, callus formation, and regeneration from oat protoplasts remains a significant challenge, requiring further research. Studies have explored DH technology in oats using anthers, only one has investigated microspores. Challenges include cell fate transition, albinos, and low efficiency. Wide hybridization, while successful in some cases, shows an overall low efficiency. The application of new breeding technologies (genome editing) is not yet thoroughly established in oats. The study by Donoso [119] stands as the lone exploration of the role of TLP8 in this context. Given that oats are hexaploid, simultaneously mutating three alleles presents a significant challenge. Moreover, it is essential to confront the challenges related to public acceptance of these technologies and establish effective communication with the public regarding the safety and advantages of such innovations.

## 7. Future Perspectives

Oat tissue culture and genetic transformation are invaluable tools in oat research programs. Overcoming biological constraints, such as genotype dependence and tissue-specific methods, will pave the way for automated transformation and enhance the versatility and throughput. To some extent, these objectives can be attained through advancements in fundamental research, which aim to uncover basic biological processes and genetic background. This becomes especially feasible with the identification of additional regeneration regulators, such as morphogenic genes. The incorporation of morphogenic genes within the protoplast system could be one of the strategies to address this challenge. Upon successful protoplast regeneration, a significant avenue emerges for protoplast fusion and transformation through the utilization of Ribonucleoprotein (RNP) complexes for genome editing and numerous additional prospects. The recent sequencing of the oat genome (*Avena sativa* L.) provides significant insights for augmenting various traits [1]. Moreover, it paves the way for implementing new genomic techniques (genome editing), aimed at identifying candidate gene sequences for targeted manipulation. DH technology represents another promising avenue for oat breeding. This technology can produce homozygous oat lines in a single generation, accelerating the process of oat breeding and enabling the development of new cultivars with improved traits. In addition to conventional double haploid (DH) production methods, there is a growing body of research focused on generating double haploid plants through the utilization of haploidy inducers [120]. Overall, these technologies offer significant potential for the development of improved oat cultivars.

## Figures and Tables

**Table 1 plants-12-03782-t001:** Comparative analysis of the environmental footprint, land, and water usage of oat milk, dairy milk, and other plant-based milk alternatives (e.g., almond, soy, and rice) [11].

Drink	Land Use (m²/L)	CO_2_-Emission (CO_2_-Eq/L)	Water Use (L/L)	Eutrophication (g/L)
Dairy milk	8.95	3.15	628.2	10.65
Soymilk	0.66	0.98	27.8	1.06
Oat milk	0.76	0.9	48.24	1.62
Almond milk	0.5	0.7	371.46	1.5
Rice milk	0.34	1.18	269.81	4.69

**Table 2 plants-12-03782-t002:** Survey of studies employing to establish tissue culture system in oats.

Scientific Name	Accessions	Explants	Callus	Regeneration	Literature
*Avena*	not specified	Stem segments	No	No	[23]
*A. sativa*	Sun II	Germinating whole seedlings	Only callus formation	No	[24]
*A. sativa*	Victory	Root system	Only callus formation	No	[25]
*A. sativa*	7 Accessions (names not specified)	Seedling hypocotyl	Not specified	No	[26]
*A. sativa*	25 Accessions (Allen, Clintford, Clintland 64, Clinton, Dai, Diana, Dodge, Froker, Garland, Goodfield, Goodland, Hulless HA14, Illinois selection, 68–1644, Jaycee, Lodi, Minnesota selections, 73137, 74125, 75249, Otter, Portage, Portal, Putnam 61, Stout, Tippecanoe, and Vicland.)	Immature Embryo, Apical meristem	Not specified	Regeneration from callus	[27]
*A. sativa*, *A. sterilis*, *A. fatua*	*A. sativa* Accessions: Lodi, Moore, Lyon, Benson, Marathon, Dal, Stout, Tippecanoe, Lang, Victorgrain, Garry, Hudson, Terra, OA338, Victory, Black Mesdag, Victoria, Selma AJ 109/5, NP3/4, Karin, Rallus, and Coolabah *A. sterilis* Accessions: PI 292549, PI 295909, PI 296274, PI 296276, PI 320846, CW346, PI 374975, Cl 8295, PI 309478, MBM, G 152, 548, TS 6893, PI 317746, PI 296255, and PI 287211 *A. fatua* Accessions: Minnesota collection 1, 11, 18, 28, 35, 80, 112, 175, 218, 313, 327, 381, 406, 415, 429, 435, 464, 471, 487, 492, 495, 498, 523, 533, 611, 662, 686, 861, 931, 1141, 1149, and 1223	Immature embryos	Regenerable-type (compact, yellowish-white, highly lobed callus)	Regeneration from callus	[28]
*A. sativa*	Park	Roots of germinating seedlings	Callus with green spots	Regeneration from callus	[29]
*A. sativa*	Park	Seeds, mesocotyls, and immature embryos	Embryogenic callus (white and opaque) Non Embryogenic callus (rough and yellow)	Regeneration from both (embryogenic and non-embryogenic) callus	[30]
*A. sativa*	Victory and Park	Axillary tiller buds	Embryogenic callus	Regeneration from callus	[31]
*A. sativa*	Victorgrain, Victoria and hybrids from the cross GAF × Victoria	Immature embryos	Callus (not specified)	Regeneration from callus	[32]
*Avena* spp.	GAF-18, GAF-30, Lodi, Park	Immature embryos, seedling mesocotyls	Embryogenic callus	Regeneration from callus	[33]
*A. sativa*	Coolabah, Cooba, Blackbutt, Mortlock, Victorgrain and HVR	Immature embryos, Leaf base explants	Compact callus with somatic mebryogenic-like structutures	Regeneration from callus	[34]
*A. sativa*	Sanna, Sang and Vital	Leaf tissues from seedlings	Somatic embryogenesis	Regeneration from callus	[35]
*A. sativa*	Prairie, Porter, Ogle and Pacer	shoot apical meristem	No callus	Direct multiple shoot formation from shoot apical meristem	[36]
*A. sativa*	Fuchs, Jumbo, Gramena, Bonus and Alfred	Leaf base	Embryogenic callus	Somatic embryo germination	[37]
*A. sativa*	Ankara-76, Ankara-84, A-803, A-804, A-805, A-821, A-822, A-823, A-824 and A-825	Mature embryo	Nodular and white to cream in color callus	Regeneration from callus	[38]
*Avena* spp.	Gaf/Park, GP-1, Garland, Park, Corbit, F2 progenis of GP-1 × Corbit and F2 progenis of Corbit × GP-1	Mature embryo	Callus with somatic embryos	Regeneration from callus	[39]
Not mentioned	Malgwiri and Samhangwiri	Mature embryos and leaf base segments	Callus (not specified)	Regeneration from callus	[40]
*A. sativa*	Meliane	Mature caryopses	Somatic embryogenesis	Somatic embryo germination	[41]
*A. sativa*	JO-1 and OS-6	Mature embryo	Callus (not specified)	Regeneration from callus	[42]
*A. sativa*	Kent	Grains	Callus (not specified)	Regeneration from callus	[43]

**Table 3 plants-12-03782-t003:** Summary of the studies used to establish genetic transformation.

Scientific Name	Accessions	Explant Type	Callus	Regeneration Type	Genetic Transformation Method	Age of the Explants	Candidate Genes Used	Selection /Marker Gene	Transgenicity Confirmation	Transgene Inheritance	Agronamic Trait of the Gene	Literature
*A. sativa*	Lines derived from GAF30/Park	Immature embryos	Friable embryogenic callus	Not specified	Bombardment	2-week-old friable and embryogenic callus	*GUS*	*PPT*, *GUS*	Southern blot	yes	None	[44]
*A. sativa*	Jumbo and Fuchs	Leaf base explants	Embryogenic callus	Somatic embryo germination	Bombardment	Leaf bases from young seedlings 1mm size	*UIDA*	*PPT*	Southern blot	Mendelian inheritance	None	[45]
*A. sativa*	Garry	Shoot meristematic cultures	No	No	Bombardment	6-month-old shoot meristematic cultures	*UIDA*	*BAR*	Southern blot	Mendelian and non-Mendelian segregation	None	[46]
*A. sativa*	Prairie, Ogle and Pacer	Shoot apical meristem	No	Direct multiple shoot formation from shoot apical meristem	Bombardment	1-month-old multiple shoot cultures	*HVA1*	*BAR*, *GUS*	Southern blot	yes	osmotic stress	[49]
*A. sativa*	Garry	Shoot meristematic cultures	No		Bombardment	4-month-old shoot meristematic cultures	*GFP*	*BAR*, *HPT* and *NPTII*	Southern blot	yes	None	[50]
*A. sativa*	Bajka, Slawko, and Akt	Immature embryos and leaf base segments	Embryogenic callus	Via somatic embryogenesis	*Agrobacterium tumefaciens* (LBA4404 pTOK233), EHA101 (pGAH), AGL1 (pDM805) and AGL1 (pGreen)			*KANAMYCIN*, *PPT*	Southern blot	yes	None	[47]
*A. sativa*	Jo-1	Leaf base explants and mature embryos	Embryogenic callus	Not specified	*A. tumefaciens* (GV3101)		*GUS*	*HPTII*	PCR	Not detected	None	[48]

**Table 4 plants-12-03782-t004:** Summary of the studies on the establishment of haploid technology.

Scientific Name	Accessions	Explants	Inductive Treatment	Callus	Regeneration	Albinos	Literature
*Avena*	Ajax, Cartier, Clintland 64, Gemini, Hinoat, Hudson, OT-184, Pendek, Q.O.58.22 (Dorval Xyamaska), Q.O.64.31 (Harmon × Wb 16385), Roxton, Black Mesdag, Cayuse 0, Clintford, Lodi, In 73231, Stout, HED-147, O.A.338, O.A.424-1, Acton, Bento, Cabot, Clinton, Dorval, Harmon, Kelsey, Lasalle M.C, Mabel, Orbit, Stornont, Tarpa, Actor, Leanda Maldwyn, Manod, Marvellous, Milford, Nelson, Palu, Selma, Yielder, Amuri, Arlington, Ausable, Bell, Bingham, Borrus, Bravo, Bruce, Colfax, Cortez, Diana, Elan, Earl haig, Early Miller, Fayette, Florida 500, Forward, Fraser, Holden, James, Legacy, Maris Osprey, Neal, Noire Precoce de Noisy, Oneida, Clintland 64 × Ajax, Clintland 64 × Q.O.58.22, Clintland 64 × Q.O.64.31, Hinoat × Clint land 64,OT-184 × C. l. 3387, OT-184 × Q.O.58.22, OT-184 × Q.O.64.31, P.I.269182 × Q.O.64.31, Q.O.S8.22 × Hinoat, Q.O.64.31 × Q.O.58.22, and Q.O.64.31 × Ajax	Anthers Ovaries	27 °C Dark 15% sucrose concentration warm 24–30 °C	yes	Regeneration from ovary culture		[73]
*A. sativa*, *A. fatua*, *A. sterilis*	Stout and Clintford	Anthers	not specified	yes	no	not specified	[74]
*A. sativa*	Clintford Stout Benson Victorgrain Garland Moore Lyon Stout/Clintford F1 Clintford/Benson F1 Clintford/Victorgrain F1 Clintford/Garland F1 Clintford/Moore F1 and Clintford/Lyon F1	Anthers	Cold pretreatment (4–8 °C)	yes	yes	not specified	[75]
*A. nuda*	Naked oat	Anthers		yes	yes	not specified	[76]
*A. sativa*, *A. nuda*, *A. byzantia*, *A. sterilis*	Stout, Puhti, OT 194, Foothill, Pazano, Fulghum and CAV 2648	Anthers	4 °C	yes	yes	yes	[77]
*A. sativa*	Stout (line WW 18019) and CAV 2648	Anthers	32 °C	yes	yes	yes	[78]
*A. sativa*, *A. sterilis*	Stout, Puhti, Sisko, Virma, Ryhti, Nasta, Sv 86432, Kolhu, WW 18019, Cascade, Heikki, Hankkijan Vouti, Yty, Roope, Veli, Aarre, Freja, Pol, Park, Titus, Katri, ME 7539, Aio, Sisu, Pegaz, STH 180, Wiesel, Maldwyn, Myriane, Avesta, Ogle, Ceal, 0T257, Talgai, Semu 4.004,Fuchs, Hja 86008, Hja 85013, Mostyn, Salo, Amby, STH 7518, Ebene, Rollo, CAV 1126, 16, 3a, CW 537, Ciav 2321, 55, CW 533, CW 453, CAV 1095, CD 7983, CAV 1191, CAV 2941, CAV 3175, WAHL 6, CAV 2057, Lisbeth, Jo 1418, Jo 1419, Bor 1335, Bor 1267 Bor 1306, Fuchs × PC 62, 80r70623 × 80r70818, Hja 88612 × Bor 70584, NS 126-93 × 1186-4189, HjaB77l2 × APR 166, Puhti × CAV 2648, and KP 9304 × CAV2648	Anthers	4 °C	yes	yes	yes	[79]
*A. sativa*, *A. sterilis*	WW 18019, Kolbu and CAV 2648	Anthers	32 °C	yes	yes	yes	[80]
*A. sativa*	Lisbeth, Virma, Cascade, Kolbu, WW18019, OT 257, Stout, Sisu, Katri, Yty, and Sisko	Anthers	32 °C	yes	yes	yes	[81]
*A. sativa*	Lisbeth, Aslak	Anthers	32 °C	yes	yes	yes	[82]
*A. sativa*	CHD: 1705/05, 1717/05, 1725/05, 1780/05, 2038/05, 1889/05, 1893/05, 1903/05, 1944/05, 1954/05, 1956/05, 1967/05, 1985/05, 1989/05 and 1997/05	Anthers	4 °C	yes	yes	yes	[83]
Not specified	Lisbeth × Bendicoot, Flämingsprofi × Rajtar, Scorpin × Deresz, Aragon × Deresz, Deresz × POB 7219/03, Bohun × Deresz, Krezus × Flamingsprofi, Krezus × POB 10440/01 and Cwal × Bohun	Anthers	4 °C	yes	yes	yes	[84]
*A. sativa*	2000QiON43 (LA9326E86)	Microspores	4 °C	yes	yes	yes	[85]
*A. sativa*	Akt, Bingo, Bajka, and Chwat	Anthers	4 °C (2 and 3 weeks) 32 °C 24 h	yes	yes	no	[86]
*A. sativa*	Bingo and Chwat	Anthers	4 °C (2 weeks) 32 °C 24 h	yes	yes	no	[87]

## Data Availability

Not applicable.

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
