# Peer review of "From Petri Dish to Field: Plant Tissue Culture and Genetic Engineering of Oats for Improved Agricultural Outcomes"

_plants, 2023, doi:10.3390/plants12213782_

Round 1

Reviewer 1 Report

Comments and Suggestions for Authors

I have thoroughly reviewed the manuscript titled " From Petri Dish to Field: Plant Tissue Culture and Genetic Engineering of Oats for Improved Agricultural Outcomes" submitted to Plants. The manuscript aligns well with the purpose of the journal and aims to address a relevant concern about climate change. This review delves into the importance of oats and their different applications, highlighting the advantages of plant cell culture and in particular it provides an overview of plant tissue culture, genetic transformation, protoplast technology, aploid technology and genome editingh, highlighting lights and shadows of each application on oats.

Suggestions for Improvement:

Keywords: in my opinion they are too many and too long, they must be of one word, at most 2 words.

However, there are several areas that warrant attention before considering the manuscript for publication. In particular, the points 3.1 Genome editing or site directed mutagenesis in oats and 3.2 CRISPR are a bit generic, which could be further integrated and deepened with more specific references, concerning oats. Also the last part 4 Future perspectives and 5. Potential Challenges need to be integrated further.

Comments on the Quality of English Language

The language is fluid and grammatically sound.

Author Response

Comment 1: Keywords: in my opinion they are too many and too long, they must be of one word, at most 2 words.

Response: 1 Changes were made as per reviewer suggestion,

Comment: 2 However, there are several areas that warrant attention before considering the manuscript for publication. In particular, the points 3.1 Genome editing or site directed mutagenesis in oats and 3.2 CRISPR are a bit generic, which could be further integrated and deepened with more specific references, concerning oats. Also the last part 4 Future perspectives and 5. Potential Challenges need to be integrated further.

Response: 2 We acknowledge that the literature on genome editing and site-directed mutagenesis in oats, particularly in the context of CRISPR technology, is indeed limited. Specifically, there is only one thesis available on this subject, resulting in a scarcity of specific references concerning oats. However, it is important to highlight that the available thesis has been thoroughly addressed and integrated into the review. Additionally, in accordance with the reviewer's suggestion, we have supplemented this information with additional relevant content to enhance the depth and specificity of the discussion regarding genome editing and CRISPR in oats. Besides this Based on the reviewer's suggestions, we have made revisions to both the future prospects and potential challenges.

Reviewer 2 Report

Comments and Suggestions for Authors

This manuscript represents a comprehensive review of tissue culture and genetic engineering research on oats. The subject is well covered but the paper organization needs to be improved and some forward-looking assumptions revisited. 

Line 55 (Introduction): Please provide a reference for "The worldwide oat milk market is projected to grow at a rate of 15% annually from 2020 to 2028." That is a bold prediction that requires specific citation. 

The title of Table 1 appears to be misleading, given the data in the table (see soya). Land use is indicated as contributing to a smaller environmental footprint whilst the sentence above the table projects that market will grow at a rate of 15% annually from 2020 to 2028 (without acknowledging this might increase 'land use'). 

Line 72 (Introduction) - Is it meant to favorably compare oats to rice and almonds instead of soy and almonds? 

Line 80 (Introduction) - also throughout the paper, be cautious with forward-looking statements regarding impact of gene editing on quantitative traits such as yield (potential? based on indirectly impacting other, qualitative traits?).

Section 2 - Whilst Table 2 lists research published on the subject in chronological order (fine), would it be better to organize section 2 into sub-sections describing different types of efforts (simple regeneration, transformation methods, etc.), whatever makes best sense for what has been done to date in oats? 'Protoplast technology' appears to be the first subheading in section 2. The one big paragraph in chronological order is difficult to follow and understand from what has been done, and from that knowledge, strategies for best way forward. This is a comprehensive review that might be followed by scientists looking for this advice.  

Why is 'Genome editing and site directed mutagenesis in oats' a subheading (3.1) under 'Doubled haploid technology' (3). Same question for 'CRISPR' (3.2). There is a wealth of information on tissue culture and genetic engineering for oats in this review paper, but it must be organized better to be useful and impactful (as it can be). Put more thought into how this review paper can be better organized - all the elements are there.

In 'Future perspectives' herbicide resistance and nutritional content are correctly identified as targets of genetic transformation, but it does not explain how these are indirect, not direct, contributors to enhanced yield and abiotic stress. Doubled haploid technology can improve the process of oat breeding, helping to improve the all-important quantitative traits so new cultivars contain good base germplasm to deliver novel gene-edited or transformed qualitative traits to market. Be careful regarding forward-looking statements that over promise on gene editing at the expense of breeding. Gene editing has promise for accelerating breeding for disease resistance (because it can be qualitative and found within the species genome) but is not mentioned in future perspectives (but is mentioned earlier in the paper).

Line 614 (Potential challenges). Why is genotype dependence in regeneration so challenging? Is an assumption made here (but not explained) that events, whether interspecies transformed or intraspecies gene edited will not be registered, protected, regulated, etc., but rather the process will be used repetitively for the same trait to create new varieties? Normally, a successful event is characterized and backcrossed into new varieties conventionally, since several events expressing the same phenotype would be difficult to manage in QA/QC seed business. This (protection/ownership of gene edited traits) may be as much a barrier as 'public acceptance' of these technologies.  Would improving the efficiency of a good regenerable cultivar (reselecting in tissue culture) be a better strategy than trying to make all cultivars more regenerable? 

Author Response

Comment:1 Line 55 (Introduction): Please provide a reference for "The worldwide oat milk market is projected to grow at a rate of 15% annually from 2020 to 2028." That is a bold prediction that requires specific citation. 

Response:1 This line has been removed from the manuscript

Comment:2 The title of Table 1 appears to be misleading, given the data in the table (see soya). Land use is indicated as contributing to a smaller environmental footprint whilst the sentence above the table projects that market will grow at a rate of 15% annually from 2020 to 2028 (without acknowledging this might increase 'land use'). 

Response:2 The title of Table 1 has been revised to: "Comparative Analysis of the Environmental Footprint, Land, and Water Usage of Oat Milk, Dairy Milk, and Other Plant-Based Milk Alternatives (e.g. Almond, Soy, and Rice)

Comment:3 Line 72 (Introduction) - Is it meant to favorably compare oats to rice and almonds instead of soy and almonds? 

Response:3 The sentences were modified to prevent any potential misinterpretations

Comment: 4 Line 80 (Introduction) - also throughout the paper, be cautious with forward-looking statements regarding impact of gene editing on quantitative traits such as yield (potential? based on indirectly impacting other, qualitative traits?).

Response: 4 Forward-looking statements were modified in accordance with the reviewer's suggestions

Comment: 5 Section 2 - Whilst Table 2 lists research published on the subject in chronological order (fine), would it be better to organize section 2 into sub-sections describing different types of efforts (simple regeneration, transformation methods, etc.), whatever makes best sense for what has been done to date in oats? 'Protoplast technology' appears to be the first subheading in section 2. The one big paragraph in chronological order is difficult to follow and understand from what has been done, and from that knowledge, strategies for best way forward. This is a comprehensive review that might be followed by scientists looking for this advice.  

Response: 5 The tissue culture and transformation section is divided into two segments.

Comment: 6 Why is 'Genome editing and site directed mutagenesis in oats' a subheading (3.1) under 'Doubled haploid technology' (3). Same question for 'CRISPR' (3.2). There is a wealth of information on tissue culture and genetic engineering for oats in this review paper, but it must be organized better to be useful and impactful (as it can be). Put more thought into how this review paper can be better organized - all the elements are there.

Response: 6 The sections were reorganized in accordance with the reviewer's suggestions.

Commnet: 7 In 'Future perspectives' herbicide resistance and nutritional content are correctly identified as targets of genetic transformation, but it does not explain how these are indirect, not direct, contributors to enhanced yield and abiotic stress. Doubled haploid technology can improve the process of oat breeding, helping to improve the all-important quantitative traits so new cultivars contain good base germplasm to deliver novel gene-edited or transformed qualitative traits to market. Be careful regarding forward-looking statements that over promise on gene editing at the expense of breeding. Gene editing has promise for accelerating breeding for disease resistance (because it can be qualitative and found within the species genome) but is not mentioned in future perspectives (but is mentioned earlier in the paper).

Response: 7 Based on the reviewer's suggestions, we have made revisions in the future perspectives section

Comment: 8: Line 614 (Potential challenges). Why is genotype dependence in regeneration so challenging? Is an assumption made here (but not explained) that events, whether interspecies transformed or intraspecies gene edited will not be registered, protected, regulated, etc., but rather the process will be used repetitively for the same trait to create new varieties? Normally, a successful event is characterized and backcrossed into new varieties conventionally, since several events expressing the same phenotype would be difficult to manage in QA/QC seed business. This (protection/ownership of gene edited traits) may be as much a barrier as 'public acceptance' of these technologies.  Would improving the efficiency of a good regenerable cultivar (reselecting in tissue culture) be a better strategy than trying to make all cultivars more regenerable? 

Response: 8 Based on the reviewer's suggestions, we have made revisions in the Potential challenges section

Round 2

Reviewer 1 Report

Comments and Suggestions for Authors

The review has been improved: in its present formt is clear, complete and I think it is of relevance to the field. Paragraph 5.1 CRISP has been improved.

Also the 6. Potential challenges and 7. Future perspectives are good!